# Get-togethers: Guided Peer-Support Groups for Young Carers

**DOI:** 10.3390/healthcare12050582

**Published:** 2024-03-02

**Authors:** Eva Schellenberg, Rosa M. S. Visscher, Agnes Leu, Elena Guggiari, Sarah Rabhi-Sidler

**Affiliations:** 1Careum School of Health, Kalaidos University of Applied Sciences, 8006 Zürich, Switzerland; eva.schellenberg@careum-hochschule.ch (E.S.); rosa.visscher@careum-hochschule.ch (R.M.S.V.); elena.guggiari@careum-hochschule.ch (E.G.); 2Institute for Biomedical Ethics, Medical Faculty, University of Basel, 4056 Basel, Switzerland; agnes.leu@unibas.ch

**Keywords:** young carers, peer-support, implementation, evaluation, social network, inclusion, participation, life skills

## Abstract

To address Young Carers’ (YCs) needs for space and opportunities to reflect and exchange, a guided peer-support programme, the “Get-togethers”, was developed in collaboration with YC in Switzerland in 2018. In order to evaluate if the Get-togethers were able to meet their originally set goals of (1) strengthening support among YCs, (2) promoting their life skills, (3) strengthening their social network and (4) promoting the inclusion and participation of YCs, participants of the Get-togethers were asked to complete a short questionnaire about their participation in and experiences with the Get-togethers. We also analysed the standard documentation of 17 Get-togethers held between May 2021 and September 2023. Overall, the Get-togethers were rated positively in almost all areas of the survey and the documentation, indicating that the four originally set objectives of the Get-togethers were (at least largely) achieved. The Get-togethers covered a large part of the needs of YCs, such as emotional support and opportunities to relax and exchange with people in a similar situation, yet they largely failed to reach minor YCs and male YCs. Further support programmes should be developed to address the different needs of different groups of YCs.

## 1. Introduction

In Switzerland, around 8% of children and teenagers aged 9 to 16 years are Young Carers (YCs), i.e., they regularly look after and/or care for one or more close person(s) with an illness or impairment [1,2]. National and international studies have found a lower quality of life and less social participation among YCs compared to their non-caregiving peers [1,3]. Many YCs remain invisible and without support [4,5,6], and they experience short- and longer-term risks in various areas of life, such as disadvantages for their own health and psychosocial well-being [7,8,9,10,11,12,13]. 

Previous work has shown that YCs lack space and opportunities to reflect on their own situations, both alone and together with other young people in a similar situation [14]. In some countries, support programmes tailored to YCs providing opportunities for social participation and reflection have been developed, e.g., the Buddy Programme in Denmark [15] or an art and respite programme in France [16]. In Switzerland, however, there were hardly any support programmes aimed at YCs [17,18], and for this reason, the Get-togethers were set up.

This report aims to evaluate to what extent the Get-togethers meet the needs of YCs. 

## 2. Materials and Methods

### 2.1. Development of the Get-togethers

To address YCs and their needs, a guided peer-support programme was developed in collaboration with YCs. These so-called Get-togethers are the first support programme specifically tailored to YCs in Switzerland and aim to achieve four main goals (Table 1). Get-togethers are low-threshold exchange meetings for and with YCs and Young Adult Carers (YACs) aged 15 to 25, as well as for former YCs. The Get-togethers offer young people the opportunity to exchange ideas and experiences and to undertake fun activities with a group of peers under the guidance of a trained facilitator. The recruitment of participants mainly took place via professionals in education, healthcare and social services who have contact with YCs in their daily work and via social media. Since the end of 2018, these Get-togethers (at no cost to the participants) have taken place 6–8 times a year (on-site and online). After a pilot phase from December 2018 until February 2021, the Get-togethers have taken place within the three-year project “Switzerland-wide Support Programmes for Young Carers” since March 2021.

### 2.2. Participants

All YCs, YACs, and former YCs who contacted our university regarding the Get-togethers and consented to being informed about our activities between the end of 2018 and March 2023 were asked to participate in this preliminary evaluation. They had to be between 15 and 35 years old and have a sufficient understanding of German. The evaluation, conducted in May 2023, received approval in April 2023 from the local research ethics committee of the Kalaidos University of Applied Sciences.

### 2.3. Data Collection 

To examine whether the Get-togethers achieved the project goals outlined in Table 1, we developed an online survey. In order to keep the questionnaire (Table A1 in Appendix A) as short as possible (max. 5 min) to obtain a high response rate, 12 of the 21 indicators were represented by questions included in the online questionnaire. Participants had to indicate to what extent they agreed on a 5-point Likert scale (1: do not agree at all, and 5: agree completely) to a list of statements. Furthermore, it comprised questions regarding participation and sociodemographic information (age, gender, place of residence), and respondents had the possibility to provide free-text comments at the end of the survey.

In addition, the Get-togethers were documented by the group leader after each meeting by recording key data (number of registrations and cancellations, duration, etc.), discussed topics and observations.

All participants received an email with a link to the online questionnaire created with Qualtrics XM software Version Mai-June 2023 (Qualtrics International Inc., Provo, UT, USA). A reminder was sent out two weeks after the original email. 

### 2.4. Data Analysis

To evaluate if the goals of the Get-togethers were met, frequency distributions were extracted from the questionnaire responses; only fully completed questionnaires were considered for analysis. Data from the open-ended questions (e.g., positive and difficult experiences) were subjected to a thematic content analysis to allow for their grouping into themes. 

Since goal 2 (promoting life skills) and goal 4 (promoting inclusion and participation) were not fully covered by the survey, we analysed the Get-together documentation with regard to the indicators (2.3/4.2/4.3/4.4/4.5) of these goals and to interpret the unexpected results of the survey. This was also carried out by means of frequency distributions and a thematic content analysis. No further statistical tests were performed.

## 3. Results

### 3.1. Respondents

Thirty-nine Y(A)Cs and former YCs received an invitation to participate in the study. In total, 14 participants completed the questionnaire, whilst 2 participants did not complete it in full, leading to a response rate of 36% (14 out of 39). All of the participants were female and aged between 18 and 31 years, with a median age of 23 years. Half of the participants (n = 7) took part in at least one Get-together (Figure 1), of whom six took part in Zürich and one in Basel. The other 7 of the 14 respondents never attended a Get-together. The most frequent reasons for non-participation were a lack of time and a lack of interest (n = 7; multiple answers were possible).

### 3.2. Goal 1: Strengthening Social Support

Half of the participants attended the meetings regularly (Table 1, indicator 1.1/3.3) and were present at about every second meeting. Figure 2 shows how all the participants agreed that it is good that the Get-togethers exist (1.2) and that the exchange within the group is helpful for them (1.3). Likewise, they all completely agreed that the atmosphere at the meetings is positive (1.4/3.2). All the participants felt rather or completely understood (1.5) and supported by the group (1.6), and the majority also felt encouraged by the group (1.7).

### 3.3. Goal 2: Promotion of Life Skills

All the participants agreed that they shared personal experiences in the group (indicator 1.7/2.1). Most of them agreed that they got useful advice from other participants (2.2). 

At virtually all the meetings, the participants reported that they had fun (2.3). They were happy to see each other again or to meet new participants. Likewise, they enjoyed the joint activities, such as a guided tour of the city, a game, or a visit to an event.

### 3.4. Goal 3: Strengthen the Social Network of Young Carers

Most of the participants also agreed that they made new contacts (indicator 3.1). As there were often new participants at the Get-togethers, additional connections were possible over time.

About half of the participants did not agree with the statement that they had regular contact with people outside of the meetings (3.4/3.5). However, the analysis of the documentation showed that the YCs have created a WhatsApp group and invited new participants to it. 

### 3.5. Goal 4: Promotion of the Inclusion and Participation of Young Carers

Goal 4 was evaluated predominantly based on the documentation, except for indicators 4.1 and 4.6, which were partially covered by the questionnaire. The analysis showed that the YCs did contribute their ideas for their preferred day, time, place, duration or activities for the Get-togethers (indicator 4.3). The support role, difficulties and possible ways of dealing with them were a recurring topic at the meetings between the participants (indicator 4.4). The participants also showed great interest in participating in activities to raise awareness for YCs (indicator 4.5) by participating in conferences, presentations, panel discussions, etc. Furthermore, two participants launched a podcast on the topic of YCs (4.5). The professional guidance was appreciated by the majority (4.6).

Reaching target groups that are difficult to access and having a diverse group at the Get-togethers (indicators 4.1/4.2) was partially achieved. The Get-togethers reached only YACs (but no minors) and former YCs, almost exclusively female ones. Some male YCs were reached, but their participation in the Get-togethers is still low: only one young man participated, but several times.

## 4. Discussion

Overall, the Get-togethers were rated positively with respect to the four goals defined to meet the needs of YCs. Out of 21 indicators, 15 were confirmed in the survey and 4 in the documentation, indicating that the four objectives of the Get-togethers were (at least largely) achieved. 

Reaching diverse groups of YCs and giving them the opportunity to participate in the Get-togethers (indicators 4.1/4.2) seems more difficult to achieve since no minor YCs and only a few male YCs could be reached. Exclusively female YCs completed the survey, which limits the interpretation of the results to females only. In other words, these two indicators were only partially confirmed and constitute a limitation of this study.

Two of the indicators were assessed differently by the two sources. One reason for the different assessments of indicators 3.4 (“Participants exchange contact information on their own initiative, network with each other via chat if necessary”) and 3.5 (“Contacts are maintained inside and outside the Get-togethers”) could be that the YCs maintain contact outside the meetings, but rather irregularly, or that they would like to have more frequent contact than they do currently.

Although there are some programmes and activities that can be supportive for YCs in Europe, they do not always target YCs in particular [17]. In addition, there are only a few published evaluations of support programmes for YCs worldwide. In France, the national association JADE offers an art and respite programme for YCs [16]. An evaluation of this programme showed a significant increase in the quality of life of the participants. These results indicate that offers over a limited period of time are helpful in a similar way as the recurring Get-togethers taking place over several years. This programme also reaches mostly female YCs, just like the Get-togethers.The Buddy Programme in Denmark [15] aims to provide support to YCs aged 5–15 years. Over a period of several months, YCs are assigned a buddy (a volunteer student) to engage in fun activities together. The findings of this study show the different needs of minor YCs, who are addressed by this programme.

So far, there does not seem to be one programme that appeals to all YCs at the same time. Depending on the type of programme, different groups of YCs are reached. This could be due to the type of activity, or it is possible that certain groups are not reached by the advertising/communication or that some do not identify as YCs and therefore do not feel addressed. In the UK, a variety of support programmes for YCs have been set up over the last few years, and some of them have been evaluated [19]. The evaluation revealed the individual circumstances of YCs, which result in a range of different needs, which in turn require different types of support programmes. The goals are to enhance the personal and interpersonal resources of YCs, mainly through support in schools and the offer of a range of respite activities. 

These evaluations from three different countries show that different types of support programmes can lead to similar results regarding the improvement in the situation of YCs. With different types of support programmes, more YCs can be addressed, e.g., different age groups and genders, YCs with different backgrounds or those with different intensities of caring responsibilities.

The Get-togethers mainly reach young women. They seem to be attracted by the activities on offer and the opportunity to socialise. Over the years, a core group of YCs and former YCs has formed who have gotten to know each other well. This has resulted in a very personal exchange at the meetings, where people asked each other for tips and help with private problems, for example. This is a major advantage of such a long-term programme. The core group comprised a fairly wide age range (18–35 years). This mixed age group resulted in mentoring by the older (former YCs) for the younger YCs.

The professional guidance of the group (which was also rated positively in the evaluation) is another benefit of the Get-togethers. The participants do not have to deal with the organisation and moderation of the meetings; this is all carried out by the leader. This is conducive to a stress-free process and dialogue. The voluntariness of participation and the possibility to withdraw, sign in or out of a meeting at any time, even at very short notice, or take a break at any time are also made possible through professional guidance and are considered success factors for the Get-togethers. A great advantage is also seen in the flexible and needs-oriented settings of the Get-togethers.

The Get-togethers are recognized as a “model of good practice” for carers by the Swiss federal office of public health [20]. Furthermore, various institutions showed interest in setting up a similar opportunity in other regions of Switzerland. But although there is a need for sharing experiences among YCs, not all YCs can be reached with the current set-up of the Get-togethers. In order to reach all YCs, adaptations to the Get-togethers are needed, for example, for male YCs. A study on male YCs in Switzerland points out that traditional gender images can prevent awareness and identification as YCs, which means it could be harder for male YCs to identify themselves as such. This could be one important reason why this group is very difficult to reach. With regard to the Get-togethers, it is suggested that a focus on activities and resources is important for boys and young men [21]. It may be worth considering if a Get-together with a stronger focus on activities or with other types of activities may be a more suitable approach in order to better reach male YCs. As the results from the UK indicate, further opportunities should also be considered to reach minor YCs, e.g., a buddy programme and direct support in schools. 

## 5. Conclusions

The evaluation shows that the Get-togethers cover a wide range of the existing needs of YCs, such as emotional support, opportunities to exchange ideas with people in similar situations and opportunities to relax and have fun. However, the Get-togethers failed to reach all YCs, primarily minor YCs and male YCs. Further support programmes should be developed to address the various needs of different groups of YCs. Further research on the structure and implementation of programmes to meet the different needs of YCs is necessary.

## Figures and Tables

**Figure 1 healthcare-12-00582-f001:**
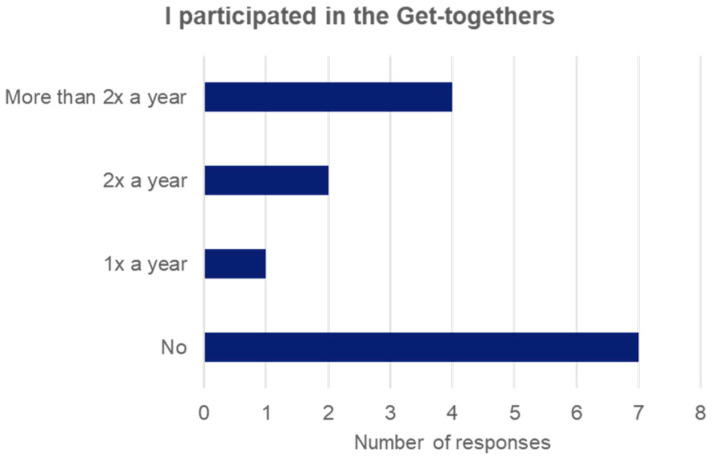
Overview of Get-together participation frequency.

**Figure 2 healthcare-12-00582-f002:**
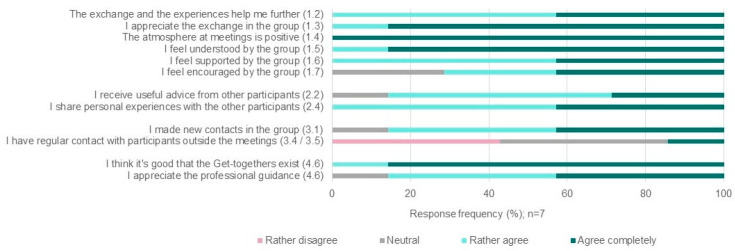
Participants’ responses to the survey questions. From the 14 complete responses, 7 individuals indicated that they participated in at least one Get-together meeting. The numbers in brackets in the legend correspond to the indicators.

**Table 1 healthcare-12-00582-t001:** Originally set goals of the Get-togethers and respective indicators (behavioural level).

Nr	Goals	Indicators
1.	Strengthening social support among Young Carers. The Get-togethers offer Young Carers the possibility to exchange views with young people in a similar situation and to support each other. This fosters a sense of mutual understanding.	1.1 Young Carers attend regularly. 1.2 Young Carers express that they find the offer useful. 1.3 The participating Young Carers appreciate the exchange in the group. 1.4 The atmosphere is considered positive. 1.5 The participants feel understood by the group and less alone. 1.6 There is mutual social and emotional support. 1.7 They actively use the setting to share personal experiences, advice and for mutual listening and encouragement.
2.	Promotion of life skills of Young Carers. Young Carers feel more confident in dealing with difficult situations and are better equipped to deal with future challenges as a result of participating in the Get-togethers.	2.1 Young Carers share personal experiences of success with the group. 2.2 Current challenges in the lives of the Young Carers are discussed in the group, and tips and experiences are exchanged. 2.3 The group has fun doing something that the Young Carers enjoy.
3.	Strengthening the social network of Young Carers. Young carers get to know other young people in a similar situation through the Get-togethers, establish contacts and strengthen their resources in the sense of social capital.	3.1 New contacts among young people were made. 3.2 The atmosphere is considered positive. 3.3 Formation of a “constant” group and regular participations. 3.4 Participants exchange contact information on their own initiative and network with each other via chat if necessary. 3.5 Contacts are maintained inside and outside of the Get-togethers.
4.	Promotion of the inclusion and participation of Young Carers. The Get-togethers provide Young Carers with space to explore their role. Young Carers are involved in decision-making processes and actively contribute their ideas and suggestions.	4.1 A diverse group of Young carers throughout Switzerland participated in the Get-togethers. 4.2 It is also possible to reach target groups that are difficult to reach and to give them the opportunity to participate in the Get-togethers. 4.3 The participating Young Carers contribute their ideas and suggestions for the Get-togethers and actively shape them. 4.4 If desired, the participants use the space to discuss their support role and reflect on it together. 4.5 If desired, the Young Carers use the space for activities to raise awareness on the topic and to get involved in the situation of other Young Carers. 4.6 The participants appreciate the opportunity to exchange ideas with other Young Carers and the professional support.

## Data Availability

The data presented in this paper are available as appropriate on request from the corresponding author. The original questionnaire in German is available on request as well.

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
