# Peer review of "Get-togethers: Guided Peer-Support Groups for Young Carers"

_healthcare, 2024, doi:10.3390/healthcare12050582_

Round 1

Reviewer 1 Report

Comments and Suggestions for Authors

TOPIC:

 Get-togethers: guided peer-support groups for Young Carers

Thank you for inviting me to review this article. My comments are as follows:

TITLE

The title is satisfactory and represents the whole article.

ABSTRACT

The abstract's components are expressed explicitly.

INTRODUCTION

The introduction is satisfactory.

METHODOLOGY

Data analysis:

It is important to be explicit about the data analysis process.

DISCUSSION

Overall, the discussion was fruitful.

CONCLUSION

The article should have a conclusion written at the end.

Reviewer 2 Report

Comments and Suggestions for Authors

This is an interesting paper with an important outlook on what still needs to be done in the area of support for young carers. Thank you for your work!

You can explore the gender aspect in more depth. Since only one male young carer is taking part, it would be:

1. important to explain the figures between male and female Young Carers (2/3 female YC?)
2. to consider and reflect why this is the case and
3. to consider and explain how this imbalance could be changed in more concrete terms

"Informed Consent Statement: Informed consent was not required." Why is informed consent not required? There are personal data.

Reviewer 3 Report

Comments and Suggestions for Authors

Dear Authors,

Congratulations on selecting a highly pertinent topic for your manuscript "Get-togethers: guided peer-support groups for Young Carers." The importance of supporting young carers, who play a crucial role in healthcare yet often remain overlooked, cannot be overstated.

To enhance the quality of this Brief Report, I recommend considering the following suggestions:

Provide more detailed descriptions of the procedures for recruiting participants and conducting the peer-support groups to enhance reproducibility and clarity for the readers.

While the results are clearly presented, elaborating on the statistical methods used for data analysis could strengthen the scientific soundness of the manuscript/Brief Report.

Discussing any limitations encountered during the study and suggesting areas for future research could provide a more balanced view and encourage ongoing exploration in this area.

Best regards
